# Ratings of the Effectiveness of 13 Therapeutic Diets for Autism Spectrum Disorder: Results of a National Survey

**DOI:** 10.3390/jpm13101448

**Published:** 2023-09-29

**Authors:** Julie S. Matthews, James B. Adams

**Affiliations:** 1College of Health Solutions, Arizona State University, Phoenix, AZ 85004, USA; julie@nourishinghope.com; 2School of Engineering of Matter, Transport and Energy, Arizona State University, Tempe, AZ 85287, USA

**Keywords:** autism, autism spectrum disorder, diet, therapeutic diets, personalized nutrition, gluten-free casein-free diet, ketogenic diet, Feingold diet, healthy diet, survey

## Abstract

This study presents the results of the effectiveness of 13 therapeutic diets for autism spectrum disorder from 818 participants of a national survey, including benefits, adverse effects, and symptom improvements. The average Overall Benefit of diets was 2.36 (0 = no benefit, 4 = great benefit), which was substantially higher than for nutraceuticals (1.59/4.0) and psychiatric/seizure medications (1.39/4.0), *p* < 0.001. The average Overall Adverse Effects of diets was significantly lower than psychiatric/seizure medications (0.10 vs. 0.93, *p* < 0.001) and similar to nutraceuticals (0.16). Autism severity decreased slightly over time in participants who used diet vs. increasing slightly in those that did not (*p* < 0.001). Healthy and Feingold diets were the two top-rated diets by Overall Benefit; the ketogenic diet was the highest for nine symptoms (though had fewer respondents); and the gluten-free/casein-free diet was among the top for overall symptom improvements. Different diets were reported to affect different symptoms, suggesting that an individual’s symptoms could be used to guide which diet(s) may be the most effective. The results suggest that therapeutic diets can be safe and effective interventions for improving some ASD-related symptoms with few adverse effects. We recommend therapeutic diets that include healthy foods and exclude problematic foods. Therapeutic diets are inexpensive treatments that we recommend for consideration by most people with ASD.

## 1. Introduction

Autism spectrum disorder (ASD) is a complex neurodevelopmental disorder involving deficits in communication, behavior, and social interaction that affects 1 in 44 children, and 4.2 times as many boys than girls [1]. ASD often involves many co-occurring symptoms, including intellectual disability, seizures, sleep disorders, gastrointestinal disorders, feeding disorders, and mood disorders The lifetime cost of caring for a child with ASD in the United States is $1.4–2.4 million (for those without and with an intellectual disability, respectively) [2]. Current treatment options include behavioral therapy [3], special education and other therapies [4], psychiatric/seizure medications [5], nutraceuticals [6], gastrointestinal treatments [7], and therapeutic diets [6,8].

Of the various treatment options, therapeutic diets have received relatively little research despite being widely used by autism families [9]. Several survey studies have reported benefits from therapeutic diets in some ASD symptoms, including behavior, communication, and health, as well as gastrointestinal issues, attention, communication, and socialization [10,11]. Clinical experience also demonstrates that dietary intervention can improve some core ASD symptoms. Case reports describe the benefits of therapeutic diets for children with ASD, including improvements in eye contact, communication, constipation, and vomiting from a gluten-free casein-free (GFCF) diet [12], as well as improvements in cognition, autism symptoms, and even a reduction in seizures from a gluten-free casein-free ketogenic diet [13].

One of the major reasons for implementing therapeutic diets is that children with ASD often exhibit self-limited diets, and may eat only a small number of foods, which increases the risk of macronutrient and micronutrient deficiencies [14]. A meta-analysis showed that children with autism spectrum disorder consume a diet lower in calcium, vitamin D, thiamine, riboflavin, vitamin B12, selenium, phosphorus, and omega-3 than typically developing control children, and calcium and vitamin D were under the recommended daily intake [15]. Additionally, an observational study found children with ASD were significantly lower than neurotypical controls in levels of calcium, magnesium, carotenes, vitamin B5, vitamin E, biotin, and lithium, with 7–31% of children below the reference ranges for these nutrients [16].

Another major reason to consider therapeutic diets is that a meta-analysis of eight studies reported that children with ASD have many differences in their gut bacteria compared to typical children [17], and many studies have reported elevated levels of intestinal yeast in children with ASD compared to controls [18]. Both bacteria [19] and yeast [20] are largely influenced by diet, and research shows the effects of food and therapeutic diets on the microbiota in mental health disorders [21] including autism [22]. Some therapeutic diets are designed to alter the gut microbiome, and it is suggested by researchers that they can be used to influence the microbiome and be customized based on personal underlying factors and needs [23].

ASD involves many more comorbidities and underlying factors for which dietary intervention may be helpful, such as mitochondrial dysfunction [24], gastrointestinal issues, immune system dysregulation, seizures, and sleep disturbances, as well as psychological conditions including anxiety, depression and behavior problems [3]. Further abnormalities include oxidative stress [25] disordered methylation, sulfation, and transsulfuration [26], and disordered oxalate metabolism [27].

Most research on therapeutic diets in ASD has focused on gluten-free casein-free (GFCF) diets, which are suggested because gluten and casein (from milk products) are common allergens [28]; several studies have reported decreased levels of lactase (needed to digest lactose in milk) in children with ASD [29,30,31,32], and there is a hypothesis that undigested casein may have an opioid-like effect in the brain [33]. The research on the effectiveness of GFCF diets is mixed, but a meta-analysis of eight randomized controlled trials of the GFCF diet found that it significantly improved stereotypical behavior of autism (5 studies) and cognitive function (3 studies) [34].

The ketogenic diet has also been found to be beneficial and improve ASD symptoms [35]. Ketogenic diets are low carbohydrate, high fat, and adequate protein diets that are widely used for seizure control for epilepsy [36], and about 30% of people with ASD eventually develop seizures, and about 80% have subclinical seizures [37]. A study on the ketogenic diet versus the GFCF diet in ASD found both diets were beneficial and provided symptom improvements, although they had benefits in different symptoms [38]. The Paleo diet and Specific Carbohydrate Diet (SCD) are additional diets that eliminate particular carbohydrates including grains and starches. The Paleo diet also avoids all legumes, dairy, and most added sugars, while SCD avoids disaccharides and polysaccharides including sucrose and lactose. Part of the reason for the use of these diets is that children and adults with ASD often have low levels of disaccharidase enzymes, contributing to dysbiosis and digestive distress [32]. Research on SCD shows it is beneficial for children with ASD by reducing autism symptoms [39].

Many other diets are commonly used for individuals with ASD, but there is less research on them. Most diets focus on removal of specific foods and compounds that negatively affect biochemistry and symptoms. The Feingold diet reduces artificial food additives and salicylates and is recommended because these substances are difficult to detoxify in certain children, contributing to hyperactivity and negative behaviors [40]. A corn-free diet, soy-free diet, or other food-avoidance diet (based on IgG or IgE food testing) removes the problematic proteins from the diet that can cause inflammation and intestinal permeability [41]. A low sugar diet is a diet low in added sugar and overall sugar intake and is beneficial as there is no nutritional need for sugar and it can contribute to overweight, digestive symptoms, poor growth, dental caries, and increase the risk of developing type 2 diabetes and cardiometabolic conditions [42], and also increase the risk of intestinal yeast infections, which are common in children with ASD [18,43]. A food-avoidance diet (based on observation) removes any foods that have been known to cause a reaction. A low oxalate diet has been proposed because impaired oxalate metabolism is an underlying factor in some individuals with ASD [27]. Many review papers examine the use of therapeutic diets for ASD, and most discuss the use of a targeted or personalized nutrition approach to dietary intervention, and include the following diets: the GFCF diet [44,45,46,47,48], ketogenic diet [45,46,47,48], Specific Carbohydrate Diet [45,47,48], low sugar diet [44,45], Low FODMAPs diet [45], elimination diet [48], low oxalate diet [48], Feingold diet [45,46], and the Mediterranean diet [47].

Other effective diet strategies that have been reported include focusing on a healthy diet by adding nutritious foods and removing unhealthy foods such as sugar and food additives [46]. A randomized controlled study of a comprehensive diet and nutrition approach, which included a gluten-free, casein-free, and soy-free diet, focusing on healthy foods along with nutrient supplementation for individuals with ASD, resulted in improved non-verbal IQ, developmental age, and many symptoms and comorbidities of autism spectrum disorder, such as language, sociability, anxiety, ritual behaviors, gastrointestinal distress, and more [49].

Many of the diets discussed above have had little research on them, and there is almost no research to compare the efficacy of one diet vs. another. Survey research on therapeutic diets for ASD provides initial evidence on the benefit and effects of many diet strategies [9,10,11]. However, some studies are rather small with 37 respondents and report on the improvement from therapeutic diets as a whole (not individual diets) [10], and other surveys are limited to results on only one or a few diets [9,11]. One huge survey study included data from over 27,000 families [50] on many treatments, including several diets, and reported 45–71% of individuals improved on various diets with rare adverse effects (2–7%), but did not specify which symptoms improved.

This paper reports on the results of a national survey of the effectiveness of 13 therapeutic diets for ASD and their associated symptom changes. This study is part of a larger survey that included nutraceuticals [51], psychiatric/seizure medications [52], and therapies, and involves 818 participants that reported on the effect of one or more therapeutic diets. It includes information on the Overall Benefits and Overall Adverse Effects, as well as specific symptoms affected.

The study was designed to obtain an understanding of the benefits and adverse effects of therapeutic diets for individuals with autism spectrum disorder, as rated by caregivers of children and adults with ASD (and some individuals with ASD). The research questions were, will therapeutic diets offer Overall Benefit and symptom improvements for individuals with ASD, and will different diets help different symptoms?

## 2. Materials and Methods

This observational study was a cross-sectional study design that reported on the results of an online survey entitled, “National Survey on Treatment Effectiveness for Autism”. The survey gathered data on the effectiveness of therapeutic diets, nutraceuticals, medications, and therapies for ASD. This paper is focused on the results for therapeutic diets.

It utilized two rating scales, one for benefits and one for adverse effects. The survey also captured data on what percentage of participants reported changes in various beneficial and adverse symptoms related to each diet.

The study design, survey, and ads for the study were approved by the Institutional Review Board (IRB) of Arizona State University. The details of the creation, data collection, and distribution of this survey can be found in a previously published study using this survey [52].

The inclusion criteria were participants in the study, parents and caregivers of children and adults with autism, autism spectrum disorder, Asperger’s syndrome, high-functioning autism, pervasive developmental disorder not otherwise specified, as well as individuals with autism spectrum disorder. Since participation was anonymous, diagnosis was not verified. Exclusion criteria were non-English speakers.

The survey had seven sections: medical history, psychiatric and seizure medication, general medication, nutraceuticals, diets, therapies, and education. This study focuses on the participants who reported the use of therapeutic diets, along with the demographics and the medical history of those participants.

The diet portion of the survey began with a question of which diets the individual with ASD had previously tried or was currently using, including a casein-free diet, corn-free diet, Feingold Diet (defined as no artificial colors, flavors, or preservatives), food-avoidance diet (based on IgG or IgE food testing), food-avoidance diet (based on observation), GAPS (Gut and Psychology Syndrome) Diet, gluten-free and casein-free (GFCF) diet, gluten-free diet, healthy diet (defined as high intake of vegetables, fruit, protein; low intake of junk food), ketogenic diet, low oxalate diet, low sugar diet, medium chain triglyceride diet, modified Atkins diet, Paleo diet, rotation diet, soy-free diet, Specific Carbohydrate Diet (SCD), other diet (with a write-in field), and none. Results are only reported here for diets with 20 or more responses.

The survey asked respondents about the Overall Benefit and Overall Adverse Effects of each diet used. There was an interval scale for rating the benefits and adverse effects of the diets. The rating scale for perceived Overall Benefit was a scale of 0–4 (0 = no benefit, 1 = slight benefit, 2 = moderate benefit, 3 = good benefit, and 4 = great benefit). The perceived Overall Adverse Effects (AE) rating scale was 0–3 (0 = no adverse effects, 1 = mild adverse effects, 2 = moderate adverse effects, and 3 = severe adverse effects). The Overall Benefit and Adverse Effects ratings of each diet were calculated as the mean of the scores of the participants. Net benefit was calculated as Overall Benefit minus Overall Adverse Effects.

The study also collected data on which symptoms changed with each diet; the list of symptom improvements and adverse effects can be found in Table 1. For each diet, the participant selected which symptom(s) were affected by the diet.

The symptom improvements for each diet were reported as the percentage of people who had improvement in that symptom with the diet. The adverse effects were reported as the percentage of people who had that adverse effect with the diet. The top symptom improvements were reported for each diet. Additionally, the diets that were highest rated for specific symptom improvements were also reported.

Comparisons on the average Overall Benefit and Overall Adverse Effects of therapeutic diets vs. nutraceuticals and therapeutic diets vs. psychiatric/seizure medications were run. The Mann–Whitney U test was used since the data was not normally distributed.

The survey also asked how strictly the participant followed the diet, as well as how much advice they received. Statistics on correlation coefficients were run between how strictly participants followed the diet and the benefits they received from the diet, between how much dietary advice they received and how strictly they followed the diet, and how much dietary advice they received and the benefits they received from the diet. Because the data was not normally distributed, the Spearman’s rank correlation coefficient or Spearman’s rho (*r_s_*) statistical test was used.

The survey gathered data on the severity of autism at 3 years of age and at the current time the survey was filled out. A value of 1–5 was given with no autistic symptoms = 1, nearly normal with only very mild symptoms = 2, mild autism = 3, moderate autism = 4, and severe autism = 5. The severity at age 3 was subtracted from current severity to calculate the change in severity. A negative number indicated a reduction in severity. The mean of change in severity was calculated for the diet users and non-diet group. Additionally, the Mann–Whitney U test was used (since data was not normally distributed) to compare any change in autism severity between participants that used dietary intervention (*n* = 486) versus those that did not (*n* = 332).

The final diet question in the survey asked “Overall, what benefit do you think diet had on your child?” This overarching question included a seven-point rating scale: the first three were much better (3), somewhat better (2), or slightly better (1); the middle response was no effect (0); and the last three were mildly worse (−1), somewhat worse (−2), or much worse (−3). The mean score was reported.

Survey data was analyzed through IBM SPSS Statistics software, version 28.0.1.1. Comparisons between groups were performed with the Mann–Whitney U test and correlations were performed with the Spearman’s rank correlation coefficient, as data were not normally distributed. A *p*-value of less than 0.05 was considered statistically significant.

## 3. Results

### 3.1. Demographics

Of the 818 participants who filled out the survey, a vast majority were primary caregivers (87%) and just over 9% were individuals with ASD. More than 50% of people with ASD were 12 years old or under, 19% were 13–18 years old and 25% were over 18 years old. Males made up 75% of the group and 25% were female. Demographic data are listed in Table 2.

### 3.2. Medical History

Of the participants, 43% had a diagnosis of autism and 24% had autism spectrum disorder, defined in the survey as less severe than an autism diagnosis (other diagnoses can be found in Table 2). The developmental history included 33% who had abnormal development from birth, with the majority (56%) having normal development followed by some form of regression and/or plateau. Those with moderate to severe autism at 3 years of age consisted of 56% of participants and decreased to 53% at the current time of the survey being conducted. There were a substantial number of antibiotics given in the first three years of life with a median of 3 rounds, 1st quartile of 1 round, and 3rd quartile of 6 rounds, where rounds were defined as “10 days = 1 round”. In total, 14% of individuals had no antibiotics from age 0–36 months, approximately 70% had 1–9 rounds, 9% had 10–29 rounds, and 6% had over 30 rounds of antibiotics by the age of 3 years old (although it is possible some of these participants confused rounds with days). Table 2 includes more details of medical history data.

### 3.3. Diets in General

Table 3 lists the Overall Benefit score for each diet, which ranged from 2.0–2.7, with an average of 2.36 across all diets. Table 3 also lists the Overall Adverse Effects (AE) scores ranging from 0–0.4 and averaged 0.1. Net benefit scores (Overall Benefit minus AE) ranged from 1.9–2.7, and these 13 diet scores were averaged for a net benefit score of 2.26. Overall Benefit and Overall Adverse Effect are also graphed in Figure 1.

A separate question was asked about the overall benefit of diet in general with a worse to better rating of −3 to +3, which resulted in a mean of 1.7 ± 1.3. This score was lower because the upper range only went to +3 vs. +4 and included negative scores for those who reported diet made symptoms worse. Since all other individual diet scores used the other benefit rating scale of 0–4, this particular score of 1.7 was not used outside of this single datapoint.

### 3.4. Benefits and Adverse Effects of the Different Therapeutic Diets

The diet with the highest net benefit was a Healthy diet with an Overall Benefit score of 2.7, followed by the Feingold diet and food avoidance diet based on IgG and IgE testing with Overall Benefit scores of 2.6. The ketogenic diet had a net benefit of 2.0 (with an Overall Benefit of 2.4 but a higher-than-average Adverse Effect score of 0.4). All of the therapeutic diet ratings are listed in Table 3.

### 3.5. Frequency of Therapeutic Diet Usage

The number of individuals that used each diet is listed in Table 4. Some participants used more than one diet. The average number of diets used per participant was 2.6 diets.

Three of the four most popular diets were gluten-free and/or casein-free related, where the most commonly diet used was the GFCF diet with 221 people; next was a healthy diet with 179 people; third was a casein-free diet with 134 responses; fourth was gluten-free with 114 participants; and the final diet above 100 users was the Low Sugar diet with 104 participants. The ketogenic diet and Paleo diet were used by only 21 respondents, so caution is needed in interpreting these results due to the small sample size. Since only diets with over 20 respondents were reported on, some diets such as the GAPS diet, low oxalate diet, and others were not included in the results.

### 3.6. Symptom Improvements from Diets

#### 3.6.1. Average Symptom Improvements from All Diets

Figure 2 shows the percentage of respondents that reported symptoms improvements due to diet, averaged across all diets. The top symptom improvements from all diets were calculated by taking the unweighted average of all diets for each symptom. General benefit received the highest percentage score (44%) but was excluded from Figure 2 to focus on specific individual symptoms, which ranged from 21% to 3%. The top symptom improvements (with average percentage of users who improved) were attention (21%), cognition (18%), irritability (18%), health (17%), hyperactivity (17%), aggression/agitation (16%), anxiety (16%), constipation (15%), diarrhea (15%), and language/communication (13%).

Since these are averages of all participants, less common symptoms in ASD, such as tics or seizures, tended to have a lower % that improved, since only a fraction of the participants have the symptom. However, some diets are chosen due to a specific symptom; for example, the ketogenic diet is used for seizures, so, presumably, more of the people using the ketogenic diet probably experience seizures.

#### 3.6.2. Symptom Improvements of the Different Therapeutic Diets

All symptom improvements are reported as the percentage of individuals who reported improvement with that symptom with a specific therapeutic diet. The top diets per symptom improvement were ranked by percentage of people who had improvement in that symptom in Figure 3, and the top five diets for each symptom are listed in Table 5. For the top five diets in Table 5, when the percentage was the same, they are ranked secondarily by which diet had a higher Overall Benefit rating. The diets that appeared in the top five were, most often, the ketogenic diet, GFCF diet, Feingold diet, food avoidance diet based on observation, and a Low Sugar diet.

The ketogenic diet was the top-rated diet for nine symptoms: attention (43%), cognition (37%), anxiety (33%), language/communication (29%), social interaction and understanding (29%), constipation (24%), seizures (19%), lethargy (19%), and depression (14%).

The Feingold diet was the top-rated diet for six symptoms: hyperactivity (45%), irritability (38%), aggression (34%), sensory sensitivity (22%), falling sleep (19%), and staying asleep (15%).

The GFCF diet was in the top two diets for cognition (29%), language/communication (25%), diarrhea (22%), social interaction and understanding (22%), sensory sensitivity (19%), and stimming/perseveration/desire for sameness 19%.

The top-rated diet for improving health was a Healthy diet (along with food avoidance based on observation) at 24%.

The corn-free diet was the best diet for diarrhea with 26% of individuals improving and second for constipation at 22%.

The Low Sugar diet was in the top two diets for hyperactivity (43%) and aggression (23%) and in third place for irritability (23%).

The Paleo diet was the top-rated diet for tics, self-injury and OCD at 10% for each.

The SCD was in the top three diets for anxiety (19%), social interaction (14%), and stimming (14%).

Table 6 shows the symptoms that improved 10% or more for each diet. Different diets were reported as beneficial for different symptoms.

### 3.7. Comparison of Diets with Nutraceuticals and Medications

The Overall Benefit scores averaging over all the therapeutic diets were compared with the average ratings for nutraceuticals [51] and psychiatric/seizure medications [52] reported previously. The average Overall Benefit of diets was 2.36, substantially higher than for nutraceuticals (1.59, *p* < 0.001), or psychiatric/seizure medications (1.39, *p* < 0.001). The average Overall Adverse Effects of diets was 0.10, similar to that for nutraceuticals (0.16), and much lower than that for psychiatric/seizure medications (0.93, *p* < 0.001).

### 3.8. Correlations between Strictly Following Diet, Advice Received, and Overall Benefit of Diet

Positive correlations were found between how strictly diets were followed and the Overall Benefit users received from the diet. The following seven diets had correlations between how strictly they were followed and the Overall Benefit rating of the diet: SCD (*r_s_* = 0.45, *p* = 0.008), Food avoidance based on observation (0.40, *p* < 0.001), GFCF (*r_s_* = 0.30, *p* < 0.001), gluten-free (*r_s_* = 0.32, *p* < 0.001), casein-free (*r_s_* = 0.26, *p* = 0.004), low sugar (*r_s_* = 0.25, *p* = 0.015) and a Healthy diet (*r_s_* = 0.17, *p* = 0.003). These correlations indicated low to moderate positive correlations that were statistically significant, and all correlations suggest that more strictly following these diets resulted in slightly better outcomes.

There were also positive correlations between the amount of advice they received and how strictly the diets were followed for Paleo, gluten-free, food avoidance based on observation, casein-free, food avoidance based on IgG/IgE, GFCF, Healthy, and Low Sugar diets. Correlations were low to moderate ranging from *r_s_* = 0.21–0.49 (*p* < 0.05).

There were low to moderate positive correlations between the amount of advice received and Overall Benefit from diet for food avoidance based on observation, gluten-free Healthy, and GFCF diets with *r_s_* ranging 0.14–0.42 (*p* < 0.05).

In summary, more advice was associated with stricter following of the diets, and stricter following of the diets was associated with slightly better outcomes.

### 3.9. Change in Autism Severity between Participants Who Used Diet vs. Those Who Did Not

Of the 818 participants in this study, 486 used therapeutic diet(s) and 332 did not use therapeutic diets. The change in autism severity between each participant at 3 years of age and at the current time of the survey was calculated for the diet group and non-diet group. Severity of autism at 3 years of age in the non-diet group was 3.33 vs. 3.50 at the current time of the survey, and severity of autism at 3 years of age in the diet group was 3.59 vs. 3.54 at the current time of the survey. The mean of change in autism severity for those that used diet was −0.05 (indicating a decrease in severity), and those that did not use diet was 0.18 (increase in severity). The Mann–Whitney U test showed that there was a significant improvement in autism severity in the diet group compared to those that did not use diet (*p* < 0.001).

## 4. Discussion

This paper highlighted the results of a survey of 18 therapeutic diets used by individuals with ASD. It reported the responses of the perceived benefits and adverse effects of the 13 diets that had over 20 responses. On average, the diets had a moderate to good benefit (2.36/4.0) with very small adverse effects (0.10/3.0).

Participants tried an average of 2.6 diets. However, the survey did not ask whether the individuals had used the diets simultaneously or not, so the study was not able to determine if some of the reported therapeutic diet results were due to diets used in combination.

Top symptom improvements reported in this study, averaging over all therapeutic diets, included attention, irritability, cognition, health, hyperactivity, aggression/agitation, anxiety, diarrhea, constipation, and language. The results were consistent with a meta-analysis of the GFCF diet for ASD, which reported significant improvements in stereotypical autism behaviors and cognition [34]. Additionally, a meta-analysis on therapeutics diets for ASD, including the GFCF, gluten-free, and ketogenic diets, found improvements in the core symptoms of autism, as well as finding them safe and effective [53]. However, most of the diets have not been formally evaluated.

The symptom improvement of “general benefit” is the highest improvement in almost all cases. This symptom improvement is very broad and unspecific and is probably more prone to the placebo effect. As such, it is difficult to interpret this response, although the higher percentage suggests that some general benefit did occur. Improvement with rare symptoms is underestimated, as not everyone doing the diet will have these symptoms. For example, around 12% of children with ASD experience seizures, reaching 26% by adolescence [54]; therefore, the effect on seizures is likely underreported for most diets, except for those that are specifically used for seizures such as the ketogenic diet.

### 4.1. Diet Surveys

This study found similar results to other survey research, which found the top areas of improvement from dietary intervention in ASD were behavior, communication, and health [10]. Three of the top nine symptom improvements were in these areas. This study also found consistent findings with another survey by Hopf et al. [9], with both studies observing that the GFCF diet was the most commonly used diet. The Hopf survey also found the low sugar, gluten-free, and Feingold diet were commonly used and had good results as did this study. However, this study had differences because it was solely focused on dietary interventions and gathered data on many more therapeutic diets. Rimland and Edelson conducted a survey of over 27,000 participants in 2009 on medications, nutraceuticals, and diets, and also found positive benefits from diet with low incidence of adverse effects, but did not specify the symptoms affected [50]. The results included the following diets and percentage of individuals who reported changes (“got better”/“got worse”) with the diet: candida diet (58%/3%), Feingold diet (58%/2%), GFCF diet (69%/3%), low oxalate diet (50%/7%), removed chocolate (52%/2%), removed eggs (45%/2%), removed milk products/dairy (55%/2%), removed sugar (52%/2%), removed wheat (55%/2%), rotation diet (55%/2%), and Specific Carbohydrate Diet (71%/7%).

### 4.2. Therapeutic Diets

#### 4.2.1. Healthy Diet

Healthy diet was defined as a diet high in the intake of vegetables, fruit, and protein and low in junk food. A Healthy diet was shown to be very beneficial with the highest net benefit (2.7) of all diets in the study, with no adverse effects reported. A Healthy diet was rated highest for improving health. A Healthy diet was also beneficial for improving constipation and symptoms of behavior and brain function such as attention, cognition, irritability, anxiety, and hyperactivity. Other research support these findings. Children with ASD were found to have a higher percentage of insufficient and unbalanced intake of healthy foods compared to typically developing children, with significantly less intake of fruit and less variety, and this was associated with lower working memory [55]. In children with ASD, a healthy diet consisting of higher consumption of vegetables, fruit, legumes, nuts and seeds and lower sugar was associated with a more beneficial gut microbiome and better GI symptom scores, while a diet with a lower intake of healthy foods was associated with a less beneficial microbiome and worse GI symptoms [56].

This poor dietary intake appears to be in part due to picky eating in children with ASD. Atypical eating behaviors were significantly greater in children with autism compared with typically developing children. Atypical eating behaviors were present in 70% of children with autism and 14.2 times more likely than in typical children [57]. Restricted food variety and food texture hypersensitivity were two of the top atypical eating patterns. In another study, food refusal was significantly higher in children with ASD with oral sensory sensitivity [58]. In a third study of children with ASD, 67% had atypical eating behaviors, 90% had one or more food intolerance as measured by IgG antibodies (to eggs, milk, and wheat), and 80% had GI symptoms [59].

While no randomized controlled trials have been performed on a Healthy diet alone for ASD, a randomized controlled trial that included a Healthy diet as part of a gluten-free, casein-free, and soy-free (GFCFSF) diet, along with substantial micronutrient supplementation, reported good results overall, and the healthy GFCFSF diet was rated the third most effective of the six treatments in the study [49].

#### 4.2.2. Feingold Diet

In this survey, the Feingold Diet was defined as a diet with no artificial colors, flavors, or preservatives. The Feingold diet is also low in naturally occurring salicylates, a food compound similar to aspirin which some people are sensitive to [60,61], found in foods such as almonds, apples, berries, grapes, orange, peaches, cucumbers, pickles, and mint.

The Feingold diet was the second highest rated diet in the study, with a net benefit score of 2.6 and no adverse effects. This study is consistent with the findings of the Hopf et al. survey, where the Feingold Diet was rated by parents higher than the GFCF diet and the Low Sugar diet [9]. The Feingold diet was rated first in six areas, particularly those related to behavior and energy regulation, including hyperactivity, irritability, aggression, sensory sensitivity, falling sleep, and staying asleep, as well as in the top three for attention, cognition, and anxiety.

In a published journal article by Dr. Ben Feingold, the creator of the diet, he stated that roughly 50% of children with hyperactivity and learning disabilities improve with the Feingold diet [40]. In his experience as a pediatric allergist, he found improvements happened in this order: hyperactivity, attention, aggression, impulsivity, writing, speech, clumsiness, cognition, and perception with the diet. This is consistent with this study, where parents reported that 45% showed improvement in hyperactivity, 34–38% showed improvements in irritability, attention, and aggression/agitation, 28% showed improvements in anxiety and cognition, and 19% or more showed improvements in sensory sensitivity and the ability to fall asleep.

A large randomized controlled trial also showed that artificial additives of preservatives and artificial colors in levels typically consumed in the diet caused significant hyperactivity in 3-year-olds and 8/9-year-olds, typically developing children, compared to placebo [62].

Salicylates are phenolic compounds, and phenols are catalyzed through sulfation by the phenol sulfotransferase (PST) enzyme requiring adequate sulfate [63]. Children with ASD are very low in sulfate and lack the ability to sulfate [16,63], so it is likely that this is a mechanism of action that makes this diet particularly beneficial.

A potential limitation in this study is the Feingold diet definition was not thorough, as it did not mention salicylates are removed in this diet. So, it is difficult to determine if participants were responding to a diet with no artificial additives or a diet that also reduced salicylates.

#### 4.2.3. Gluten and Casein Free Diets

This study separately evaluated a casein-free diet, a gluten-free diet, and a combination of a gluten-free and casein-free diet. The combined gluten-free/casein-free (GFCF) diet showed slightly higher net benefit (2.3) compared to either the casein-free diet (2.1) or gluten-free diet (1.9) alone. When looking at all symptom improvements, the GFCF diet substantially outranked the gluten-free diet and the casein-free diet individually in all symptoms. This may be because similar mechanisms of action are at play with gluten and casein. For example, research has shown high opioid compounds [33], IgG levels [28], and inflammatory markers [64] to both gluten and casein in ASD. If individuals react to both food proteins, it is likely they benefit most from removing both.

Some of the top symptom improvements for the GFCF diet in this survey were attention, cognition, language/communication, diarrhea, hyperactivity, and social interaction. The top diet for stimming/perseveration/desire for sameness was the GFCF diet. Randomized controlled trials show consistent results. While identifying high opioids from gluten and/or casein in children with ASD, researchers found significant improvements with a GFCF diet compared to controls in attention, cognition, non-verbal communication, language, aloofness, routines and rituals, learning, peer-relations, anxiety, empathy, physical contact, eye contact, sociability, sensory/motor function, and judgement of danger [65]. Children with ASD on a GFCF diet for 12 months had statistically significant improvements over baseline in communication, social interaction, inattentiveness, and hyperactivity [66]. In research that found high levels of IgG antibodies to gluten and casein in a vast majority of children with ASD, 81% of individuals improved in 3 months on a gluten-free/casein-free diet in most of the behaviors studied [28].

Results of a gluten-free only diet in children with ASD showed significant improvements in gastrointestinal symptoms and behavior compared to baseline [67], and was consistent with results from this survey. However, another study on the gluten-free diet did not find improvement in GI or ASD symptoms [68]. High-zonulin levels, which are an indication of intestinal permeability, are found in ASD, which can be caused by exposure to gluten as well as other stressors, and high zonulin was associated with a higher severity of autism [69], and may explain why some individuals find the gluten-free diet helpful. Also, consistent with this study, the gluten-free diet was found to improve social interactions in a meta-analysis of therapeutic diets for ASD; however, the meta-analysis did not find improvements in cognition and language/communication, as this study did [53]. A casein-free only diet for children with autism for 8 weeks resulted in significant improvement in behavior [70]. So, while a combined gluten-free/casein-free diet showed the best results, both individual gluten-free and casein-free diets showed improvement alone as well. The underlying factors that affect an individual likely influence whether a gluten-free and/or casein-free diet is best.

A common concern about a casein-free diet for ASD is possible calcium deficiency. Children with ASD on a GFCF diet have been found to consume significantly lower amounts of calcium than those on a regular diet [71]. However, children with ASD on a regular diet also have lower calcium levels in the body. In a study of children with ASD, where 84% were on a regular diet, the children with ASD had significantly lower levels of Red Blood Cell (RBC) calcium (14% lower) compared to neurotypical children, and 31% were below the reference range [16]. A follow-up randomized controlled trial study of a multivitamin/mineral formula with a modest amount of calcium resulted in a 43% increase in RBC calcium levels (bring it above the neurotypical control group level) [72]. Therefore, these results suggest that focusing on a healthy casein-free diet (and considering a multivitamin/mineral formula with calcium when needed) may be the best way to obtain the benefits of a casein-free diet and meet nutritional needs.

#### 4.2.4. Low Sugar Diet

The Low Sugar diet was the fourth highest rated diet with a net benefit rating of 2.4. One reason a Low Sugar diet may be beneficial is because it provides better balance of blood sugar [73]. High sugar can also cause and contribute to inflammation [74], and a mouse study shows sugar may be particularly detrimental (to glucose metabolism and cardiometabolic risk) when mitochondrial dysfunction is present [75], which is common in children with autism [24]. A Low Sugar diet was the second-best diet for reducing hyperactivity (43%, a percentage very similar to the Feingold diet) and for reducing aggression/agitation. Additional behavioral and emotional symptoms that improved on a Low Sugar diet included attention, irritability, anxiety, and cognition. This is consistent with a study on children with ASD where researchers observed an association between a higher consumption of sugar sweetened beverages and emotional problems scores [76].

A Low Sugar diet may also be beneficial because of the negative effect sugar can have on gastrointestinal health, and especially the effect on overgrowth of certain bacteria and yeast [19,20]. In a cohort study of men and women (without autism), individuals that consumed greater than one sugar sweetened beverage per day were at significantly higher risk of inflammatory bowel disease (IBD) and Crohn’s disease than those that did not consume them [77]. While these inflammatory bowel diseases are more severe than what is typically seen in ASD, gastrointestinal symptoms are common in 49% of individuals with ASD [78]. Therefore, avoiding foods that may contribute to gastrointestinal disorders, such as sugar, seem warranted, especially given all the other benefits reported. However, improvements in diarrhea and/or constipation were only found in only 8–9% of individuals on a Low Sugar diet, so improving other aspects of the diet may be more important for gastrointestinal support.

#### 4.2.5. Soy-Free, Corn-Free and Food Avoidance Diets

##### Soy-Free Diet

A soy-free diet had an Overall Benefit rating of 2.1 with no adverse effects. This diet is often implemented with a gluten-free and casein-free diet with beneficial results [49]. This is because soy can produce opioid compounds like gluten and casein [79], and soy can be inflammatory to some individuals with ASD. A study of children with ASD evaluated cytokine production to dietary proteins, and while cytokine production for soy was not significant, 7 out of 75 children with ASD and GI symptoms had elevated IFN-γ levels to soy [80]. In a study of children with autism spectrum disorder, a gluten-free, casein-free, and soy-free (GFCFSF) or casein-free and soy-free diet was implemented depending on food reactivity testing [81]. That study found that children with ASD and GI symptoms had significantly higher IL-12 (pro-inflammatory cytokine) and significantly lower IL-10 (anti-inflammatory cytokine) than those without GI symptoms, and they produced significantly less IL-10 in the unrestricted diet than the elimination diet.

While this present study only showed 2% of individuals having improvements in seizures from a soy-free diet, this is possibly due to a smaller number of people having this symptom to start. Individuals with ASD who were fed soy formula as infants were 2.6 times more likely to have febrile seizures compared to those that were not fed soy formula, 2.1 times more likely to have epilepsy, and 4 times more likely to have simple partial seizures [82]. While this study was on the development of seizures from the consumption of soy formula as infants, it may be judicious to avoid soy in cases of seizures when there are other non-dairy and non-soy substitutes available.

One concern about soy and corn products is that over 95% of soy and corn in the US is genetically modified to be more resistant to pesticides, so that higher amounts of pesticides are used on those products, resulting in higher exposure to the child [83].

##### Corn-Free Diet

There are no known studies to date on a corn-free diet for ASD. The corn-free diet was ranked in the middle for benefit with a score of 2.2 and no adverse effects. However, it scored very high for diarrhea (26% improved) and constipation (22% improved). For the symptoms of diarrhea and constipation, the corn-free diet was the top diet for diarrhea and second for constipation. One concern with corn products is that they can be contaminated with aflatoxins due to fungi growth during storage [84], as well as the concern about pesticides mentioned above.

##### Food Avoidance, Based on IgG/IgE Testing

A food avoidance diet based on IgG/IgE testing had the second highest net benefit score (2.6) along with the Feingold diet, and scored the second highest for improvements in eczema/skin problems and reflux/vomiting. Behavior and other symptoms improved including hyperactivity, irritability, attention, cognition, anxiety, sensory sensitivity, and health. Of the studies conducted on elevated IgG levels to foods in ASD, the most common improvements found were related to behavior.

In a study of children with ASD, 90% had at least one IgG food sensitivity [59]. Eggs were the top sensitivity (84% testing reactive), along with milk and wheat, and IgG antibodies were correlated with stereotyped autism behaviors. IgG antibodies to gliadin were reported in 87% of children with autism and 90% to casein [28]. In another study, anti-*α*-gliadin (AGA) and anti-deamidated *α*-gliadin IgG levels were significantly higher in children with ASD on a regular diet, and AGA-IgG was lower on the GFCF diet. Casein IgG was also significantly higher in children with ASD [85]. Researchers have used IgG testing for gluten, casein, and soy to determine which food elimination diet to implement, and found improvement in inflammatory markers when the diet was used [81]. In a previously mentioned study, researchers found high-IgG levels for casein, as well as other immunoglobulin levels for casein and proteins in dairy in children with autism at baseline, and found behavioral improvement with a casein-free diet [70].

##### Food Avoidance Diet, Based on Observation

A food avoidance diet based on observation was ranked in the middle of the group with a 2.2 net benefit rating and ranked first for reflux/vomiting. This makes sense since most individuals with ASD and/or their caregivers may notice reflux or vomiting soon after consuming a food that is problematic. An acute symptom such as this would be noticed and reported on this survey. However, there are no studies on food avoidance based on observation in ASD as this is a subjective measurement.

#### 4.2.6. Grain-Free and Carbohydrate Limiting Diets

##### Ketogenic Diet

The ketogenic diet is the second most researched diet in autism spectrum disorder after the GFCF diet. This is likely due to the clinical success in using of the ketogenic diet for pediatric seizures [86] and the increased incidence of epilepsy in ASD. Specifically, one study reported that the average prevalence of epilepsy in children with ASD between 2 and 17 years old was 12%, with the rate rising to 26% in adolescents 13 years of age and older [54]. It is also a diet used in mitochondrial disease to improve mitochondrial function [87]; therefore, with mitochondrial dysfunction occurring as a co-morbid condition in ASD, it is believed to be beneficial for these individuals.

The ketogenic diet had a good Overall Benefit score of 2.4, but a higher Overall Adverse Effects score (0.4) than most diets, resulting in a lower Net Benefit score than most diets. However, it is ranked the highest in nine symptoms, particularly those involving cognitive and brain function including attention, cognition, anxiety, language/communication, social interaction and understanding, seizures, and depression, as well as lethargy (related to mitochondrial function), and constipation.

Research shows consistent results with the findings of this survey. In a randomized controlled trial of children with ASD, the Modified Atkins Diet (a form of ketogenic diet) resulted in statistically significant improvements in Childhood Autism Rating Scale (CARS) and Autism Treatment Evaluation Checklist (ATEC) scores, with the biggest improvements in cognition, speech, and social interaction [38]. These results were consistent with the top three improvements found in this study. In another study, a modified ketogenic diet with medium chain triglycerides (MCT) over 3 months provided a statistically significant improvement in autism symptoms, and the subcategory of social affect compared to baseline [35]. In a non-randomized controlled trial of children with ASD, after 3 months on a ketogenic diet containing MCT, which was also gluten-free, researchers found statistically significant increases in ketones and acetylcarnitine (markers of improved mitochondrial function) [88].

It should be noted that the ketogenic diet is an extreme diet and should be performed under the supervision of an experienced nutritionist and/or physician.

##### Specific Carbohydrate Diet

The Specific Carbohydrate Diet had a net benefit of 2.2 (starting with an Overall Benefit of 2.4 minus 0.2 Overall Adverse Effects). SCD was ranked second in general benefit with 57% of people using the diet reporting general improvements. In total, 24% reported improvements in attention, 22% in cognition, 19% in anxiety and diarrhea, along with improvements in health, irritability, language/communication, social interaction, stimming, aggression, hyperactivity, and sensory sensitivity.

The Specific Carbohydrate Diet was first used and published by Drs. Sidney Haas and Merrill Haas in 1955 for pediatric celiac disease. In this case, the report showed that out of 191 children, 177 children had diarrhea, and 73 were able to control it within one month with SCD and all 177 by 14 months [89]. The diet focuses on the consumption of monosaccharides, and the avoidance of disaccharides and polysaccharides [89]. Disaccharides and polysaccharides require carbohydrate digesting enzymes, and researchers have found reduced levels of one or more carbohydrate digestive enzymes in 58% of children with ASD that had gastrointestinal symptoms [31]. A reduction in disaccharidase activity has been found to contribute to dysbiosis and gastrointestinal symptoms such as diarrhea in ASD [32]. This survey found the diet was beneficial for diarrhea in 19% of participants that used it. As it is unknown what percentage had diarrhea to begin with, this number likely underrepresents those that had the symptom that improved.

In a case report of a 4 year old boy with ASD with gastrointestinal problems, after he was placed on the Specific Carbohydrate Diet his gastrointestinal symptoms improved, along with his behavior and nutrient levels [90]. A non-randomized controlled trial of children with ASD on a SCD/GAPS diet plus omega 3s, ascorbyl-palmitate, probiotics, vitamin D3, and vitamin C for three months found improvements in autism symptoms [39] similar to this study. In the diet treatment group, ATEC (Autism Treatment Evaluation Checklist) scores decreased by 23% from baseline with the most significant improvements in health/behavior, socializing, irritability, and hyperactivity, as well as Parent Global Impressions—Revised 2 improved significantly in 43% of the diet group vs. 14% of the control group.

##### Paleo

While the Paleo diet was not a top ranked diet for Overall Benefit, it was the number one rated diet in three areas: OCD, self-injury and tics. Since these are less common symptoms, the effect of this diet on those symptoms may be underestimated. For example, 22% of individuals with autism have a tic disorder [91], and 10% of those on a Paleo diet had improvement in tics. This suggests that the Paleo diet may be very beneficial for tics. It is also possible that, because of the relatively small number of respondents to the Paleo diet, the results are not representative of the broader ASD population.

There is no published research known to date for the Paleo diet for ASD, but this study suggests the diet is worthy of further research given the similarities (although some differences) with other beneficial grain-free diets such as SCD, and its benefit for mental health and neurological symptoms.

### 4.3. Personalized Nutrition for a Heterogeneous Condition

This study found that many diets may be helpful in autism spectrum disorder, and different diets appear to be better for addressing different underlying factors and affecting different symptoms. Researchers believe that because ASD is a heterogeneous condition; therefore, a personalized approach to therapeutic interventions is needed to provide the most effective treatments [3,92], including personalized nutrition and diet strategies [44,45]. The best diet for the individual may depend on a variety of factors.

Additionally, when diets are used in combination, it is possible greater benefits may result. For example, people benefit from a GFCF diet more than a GF or CF diet alone, on average. Furthermore, in the case report, a gluten-free/casein-free and ketogenic diet used together was very beneficial, where the gluten-free/casein-free diet provided benefit initially, and then a ketogenic diet was added later [13]. Many other diet combinations are possible, such as combining a Healthy diet, Low Sugar diet, and any of the other exclusion diets.

Most therapeutic diets are based around the exclusion of certain foods, often based on underlying factors that may contribute to reactions to the foods or food compounds. This study also highlighted the importance of focusing on what to include in the diet, particularly healthy nutrient-rich foods. Therefore, a personalized nutrition plan should both exclude problematic foods and include healthy ones.

### 4.4. Performance of Diets vs. Nutraceuticals and Medications

Because this survey gathered information on diets, nutraceuticals, and medication with the same rating scales, these treatments were able to be compared directly to one another. Diet rated significantly higher than both nutraceutical and medications with very low adverse effects. The data also suggests that dietary interventions are widely used for individuals with ASD. Therefore, this study suggests that much more research is warranted on therapeutic diets, including studies on how to determine which diet(s) are most effective for an individual.

Diet is also an intervention that all families have access to regardless of location and with only modest resources. Even on a moderate budget, families can choose the healthiest food options they have available and can often avoid gluten- and dairy-based foods without a large expense by focusing on whole foods and avoiding expensive processed foods. Feeding their child is an empowering act parents are already responsible for on a daily basis, so parents should be encouraged by their physicians and nutritionists to explore dietary intervention. Since most physicians actually receive very little nutrition education in medical school [93], it is recommended that families find a physician who has that training and/or work with a nutritionist experienced with therapeutic diets for autism spectrum disorder.

### 4.5. Correlations of Strict Adherence to Diet and Dietary Benefit

There were statistically significant low-to-moderate positive correlations found between those who strictly followed a therapeutic diet and the Overall Benefit score of the diet. This suggests that the more strictly someone follows the diet, the better results and symptom improvement they will receive. This is significant, as this supports other research [94] and highlights a possible way to get better results with therapeutic diets. It also may suggest that the more benefit a family sees, the more strictly they follow the diet, and we suspect both are true.

There were also low-to-moderate positive correlations between the amount of advice individuals received and how strictly they followed the diets, as well as correlations between the amount of advice received and Overall Benefit from diet. In the survey, the level of advice received had four levels: no advice; limited information; some advice from a reliable source including a nutritionist, physician, book, website; and ongoing personal support from a qualified nutritionist. This suggests that the more advice an individual receives the more strictly they do the diet, and the more strictly the follow the diet the better benefits they receive. While all these correlations were not found in each of the diets, they were all present in the food avoidance based on observation, gluten-free, and Healthy diet, and all variables were not needed to have benefited from the diet. Since getting more advice and following the diet more strictly led each to better benefits, families that want to do a specific diet should be encouraged to seek out advice and follow the diet as strictly as possible.

Given that not everyone followed the diet strictly, it is possible that these Overall Benefit ratings are underestimated, and that the true benefit of the diet is higher when someone follows it strictly. Additionally, because the highest level of following the diet was considered an infraction less than once a week, this still provides a lot of potential for infractions. If there had been another level, such as infraction once per month, it is possible the correlation could have been stronger.

### 4.6. Change in Autism Severity with Therapeutic Diets

This study found there was a significant improvement in autism severity between age 3 years old and the current time of the survey compared to those that did not use dietary intervention. This data indicates that therapeutic diets can improve autism severity over time. Individuals with autism in this survey included a wide range of ages from under 3 years old to over 60 years old. Since most of these individuals were much older than 3 years old, this study shows that diet may be able to provide long term and lasting improvements.

### 4.7. Diet Therapy Is Cost Effective

Implementing a therapeutic diet is relatively low-cost compared to many therapies, such as behavior therapy, which can require several hours/day, several days/week, for years with annual costs in the tens of thousands of dollars. In contrast, diet therapies can be implemented with minimal cost, especially if preparing foods from raw ingredients instead of relying on processed foods. However, they require extra learning by the food preparer. Since most families do not have nutrition training, it is strongly recommended that families work with nutritionists who are familiar with these diets and how to best implement them. This is especially important to improve diet compliance for children who are picky eaters. Enhanced understanding of the diets is likely to result in better adherence and better effectiveness.

A nutritionist may also be able to help choose the best diet or combination of diets and aid with personalizing the dietary intervention. Finding a diet that an individual will accept and follow can be particularly challenging for people with ASD. Understanding the texture and food preferences of an individual are important, and working alongside a nutritionist can help create a diet that is more likely to be accepted. This can improve compliance with dietary intervention, which in turn may improve results.

### 4.8. ANRC Autism Treatment Rater App

The ANRC Autism Treatment Rater app is a mobile app that displays some of the data from this survey (the ratings for each diet), as well as data on many medications, nutritional supplements, and therapies. In the app, caregivers and individuals with ASD can directly compare these treatments and additional therapies to determine the best interventions based on symptoms. It is available on iPhones and can be found by searching the app store for “ANRC Autism Treatment Rater”.

### 4.9. Strengths and Limitations

Strengths of this study include the large sample size and the large number of diets surveyed. Another strength is that this study asked about specific symptom improvements with each diet; therefore, data could be gathered on whether different diets improve different symptoms. It also included a separate rating for overall adverse effects and adverse symptoms so that both benefits and adverse effects could be measured independently.

Survey studies have inherent limitations due to errors in recalling the effectiveness of the diet or the symptoms that were improved. There is also possible response bias, where those that had benefited were more likely to complete the survey. There is also a substantial placebo effect, but the comparisons between treatments should be relatively immune to the placebo effect, assuming a similar placebo effect for all treatments. Additionally, each person may implement the same diet somewhat differently or with limited compliance, which may cause some inaccuracy in the data. Because the age of participants ranges from young children to older adults, the applicability may vary for specific populations.

## 5. Conclusions

The results of this study suggest that therapeutic diets are generally safe and often effective for individuals with ASD. Therapeutic diets had significantly higher Overall Benefit than medications and nutraceuticals from the same survey, and very low adverse effects, significantly lower than medications. Additionally, individuals that used therapeutic diets had significant improvement in autism severity compared with those that did not use diet.

This survey highlights the value of therapeutic diets to address and improve the symptoms of autism spectrum disorder, as well as other common co-morbid symptoms. Dietary interventions were reported to be most effective in improving (in descending order): attention, cognition, health, hyperactivity, irritability, aggression/agitation, anxiety, constipation, diarrhea, language/communication, eczema/skin problems, stimming/perseveration/desire for sameness, sensory sensitivity, and social interaction and understanding. The highest rated diet based on Overall Benefit was a Healthy diet, and the diets that most often appeared in the top five for symptom improvements were the ketogenic diet, GFCF diet, Feingold diet, food avoidance diet based on observation, and a low sugar diet. Symptom improvement varied depending on the diet, likely due to different mechanisms of action.

Implementing a therapeutic diet is inexpensive compared to many other treatments. Hiring an expert nutritionist for one or a few sessions is inexpensive compared to, for example, years of behavioral therapy or special education, and implementing a therapeutic diet may result in better response to those therapies.

The user-friendly mobile app, ANRC Autism Treatment Rater, provides data from this study on the ratings of each therapeutic diet.

Because of the heterogeneous nature of autism spectrum disorder, a personalized nutrition approach seems to be effective, and symptoms may be used to personalize the most effective diet for the individual. Because healthy diets provided good benefit, and poor diets and nutrient deficiencies are common in ASD, a therapeutic diet should both avoid problematic foods and include nutrient-dense foods.

## Figures and Tables

**Figure 1 jpm-13-01448-f001:**
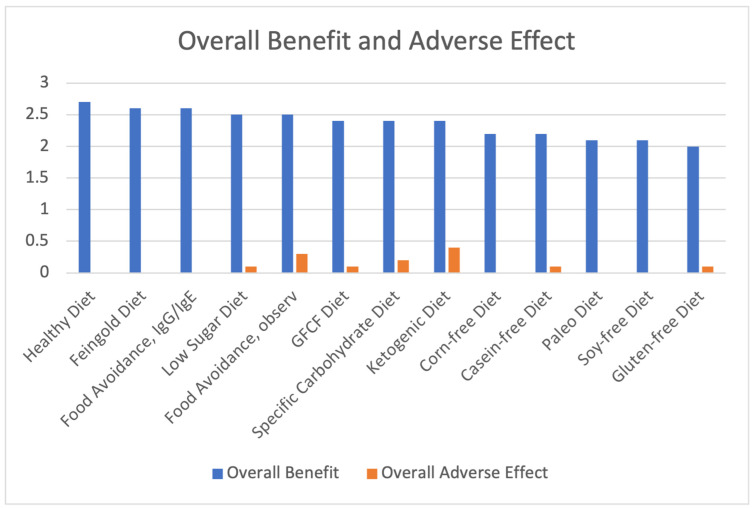
Overall Benefit and Adverse Effect of Therapeutic Diets, Sorted from Highest to Lowest Overall Benefit.

**Figure 2 jpm-13-01448-f002:**
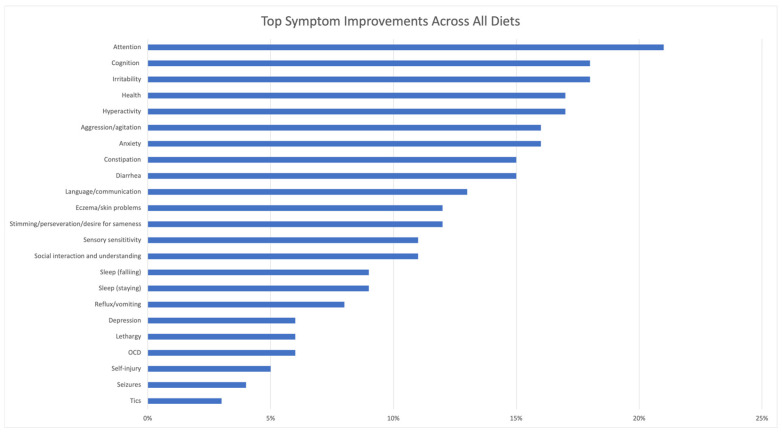
Percentage of Respondents that Reported Symptom Improvements Due to Diet, Averaged Across All Diets. General Benefits were reported by 44% of participants, but are not displayed in the graph in order to better display the other results.

**Figure 3 jpm-13-01448-f003:**
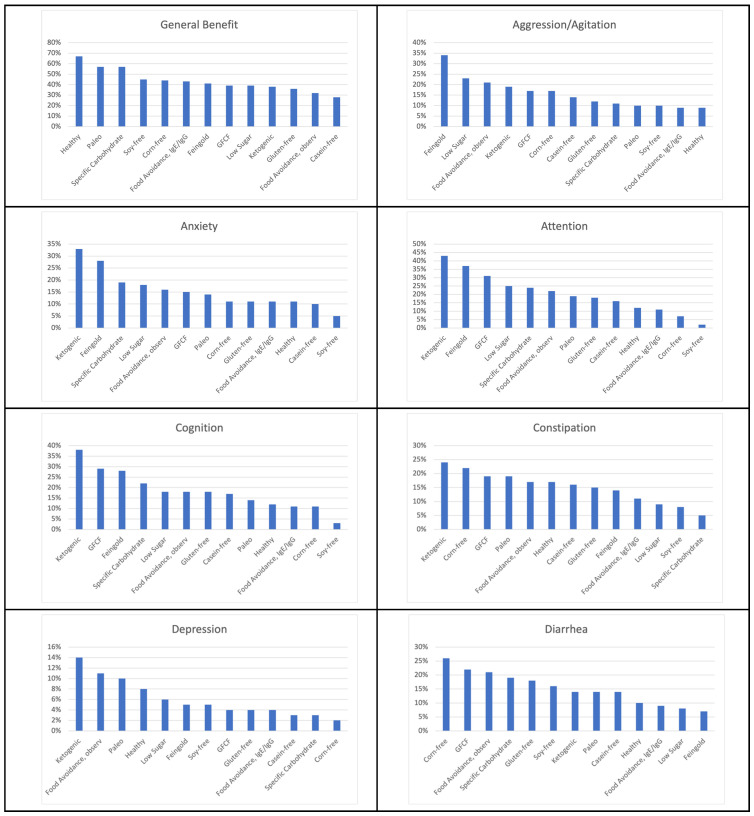
Diets Ranked by Symptom Improvement.

**Table 1 jpm-13-01448-t001:** List of Improved Symptoms and Adverse Effects.

Improved Symptoms	Adverse Effects
General benefit (no one particular symptom) Aggression/agitation Anxiety Attention Cognition (ability to think) Constipation Depression Diarrhea Eczema/skin problem Health (fewer illnesses and/or less severe illnesses) Hyperactivity Irritability Language/communication Lethargy (easily tired) OCD Reflux/vomiting Seizures Self-injury Sensory sensitivity Sleep (falling asleep) Sleep (staying asleep) Social interaction and understanding Stimming/perseveration/desire for sameness Tics/abnormal movements Other (with a write-in option)	General worsening (no one specific symptom) Aggression/agitation Anxiety Bedwetting/bladder control Behavior problems Decreased cognition (difficulty thinking/remembering) Depression Dizziness/unsteadiness Dry mouth Fatigue/drowsiness Gastrointestinal problems Headache/migraine Irritability Liver/kidney problem Loss of appetite Nausea Rash Seizures Self-injurious Sleep problems Stimming/perseveration/desire for sameness Tics/abnormal movements Weight gain Weight loss Other (with a write-in option)

**Table 2 jpm-13-01448-t002:** Demographics and Medical History.

	N	%
**Number of participants**	818	
Partipants that used dietary intervention	486	
Partipants that did not used dietary intervention	332	
**Age of individual with ASD**		
Under 3 years	21	2.6%
3–5 years	131	16%
6–9 years	181	22%
10–12 years	124	15%
13–15 years	98	12%
16–18 years	60	7.3%
19–21 years	46	5.6%
22–30 years	80	9.8%
31–40 years	30	3.7%
41–50 years	23	2.8%
51–60 years	15	1.8%
Over 60 years	8	1.0%
**Gender**		
Male	610	75%
Female	203	25%
Other	3	0.4%
**Survey filled out by**		
Primary caregiver	711	87%
High functioning individual with autism, no guardian	58	7.1%
Completed by individual with autism, has guardian	17	2.1%
Other	32	3.9%
**Diagnosis**		
Autism	349	43%
Autism Spectrum Disorder (this is less severe than a diagnosis of Autism)	199	24%
Pervasive Developmental Disorder—Not Otherwise Specified (PDDNOS)	47	5.7%
High-Functioning Autism	97	12%
Asperger’s Syndrome	119	15%
Other	7	0.9%
**Developmental history**		
Normal development, followed by major regression	169	21%
Normal development, followed by a plateau in development that lasted for several months or longer	186	23%
Normal development, followed by a major regression and a plateau lasting several months or longer	101	12%
Abnormal development from early infancy, with no major regression or plateau in development	266	33%
Other	91	11%
**Severity at age 3**		
No autistic symptoms	31	3.8%
Nearly normal, with only very mild symptoms	146	18%
Mild autism	180	22%
Moderate autism	300	37%
Severe autism	152	19%
**Severity currently**		
No autistic symptoms	4	0.5%
Nearly normal, with only very mild symptoms	123	15%
Mild autism	254	31%
Moderate autism	309	38%
Severe autism	122	15%
**Rounds of oral antibiotics from 0–36 months of age**		
Mean/Average	7	
1st Quartile	1	
Median	3	
3rd Quartile	6	
0 rounds	101	14%
1 round	135	19%
2 rounds	92	11%
3 rounds	111	16%
4 rounds	41	5.8%
5 rounds	48	6.8%
6 rounds	41	5.8%
7 rounds	15	2.1%
8 rounds	13	1.8%
9 rounds	6	0.8%
10–14 rounds	36	5.0%
15–19 rounds	13	1.8%
20–24 rounds	10	1.4%
25–29 rounds	4	0.5%
30+ rounds	45	6.1%

**Table 3 jpm-13-01448-t003:** Overall Benefit, Overall Adverse Effect, Sorted by Highest Net Benefit.

Diet	Overall Benefit	Overall Adverse Effect	Net Benefit
Healthy Diet	2.7	0	2.7
Feingold Diet	2.6	0	2.6
Food Avoidance, IgG/IgE	2.6	0	2.6
Low Sugar Diet	2.5	0.1	2.4
GFCF Diet	2.4	0.1	2.3
Food Avoidance, observation	2.5	0.3	2.2
Corn-free Diet	2.2	0	2.2
Specific Carbohydrate Diet	2.4	0.2	2.2
Casein-Free Diet	2.2	0.1	2.1
Soy-Free Diet	2.1	0	2.1
Paleo Diet	2.1	0	2.1
Ketogenic Diet	2.4	0.4	2.0
Gluten-free Diet	2.0	0.1	1.9
Average of All Diets	2.36	0.10	2.26

**Table 4 jpm-13-01448-t004:** Therapeutic Diet Frequency.

Diet	Respondents That Used Diet
GFCF Diet	221
Healthy Diet	179
Casein-Free Diet	134
Gluten-Free Diet	114
Low Sugar Diet	104
Food Avoidance, observation	82
Feingold Diet	74
Soy-Free Diet	62
Food Avoidance, IgG/IgE	54
Corn-Free Diet	46
Specific Carbohydrate Diet	37
Ketogenic Diet	21
Paleo Diet	21

**Table 5 jpm-13-01448-t005:** Summary of Top 5 Diets for Each Symptom (% of dietary users that reported that the diet improved that symptom).

Symptom Improvement	Top Diets
Aggression	Feingold (34%) Low Sugar (23%) Food Avoidance, observation (21%) Ketogenic (19%) GFCF (17%)
Anxiety	Ketogenic (33%) Feingold (28%) Specific Carbohydrate Diet (19%) Low Sugar (18%) Food Avoidance, observation (16%)
Attention	Ketogenic Diet (43%) Feingold (37%) GFCF (31%) Low Sugar (25%) Specific Carbohydrate Diet (24%)
Cognition	Ketogenic (38%) GFCF (29%) Feingold (28%) Specific Carbohydrate Diet (22%) Low Sugar (18%)
Constipation	Ketogenic (24%) Corn-free (22%) GFCF (19%) Paleo (19%) Healthy (17%)
Depression	Ketogenic (14%) Food Avoidance, observation (11%) Paleo (10%) Healthy (8%) Low Sugar (6%)
Diarrhea	Corn-free (26%) GFCF (22%) Food Avoidance, observation (21%) Specific Carbohydrate Diet (19%) Gluten-free (18%)
Eczema/skin problems	Food Avoidance, IgG/IgE (22%) Food Avoidance, observation (22%) GFCF (20%) Ketogenic (14%) Casein-free (13%)
Health	Healthy (24%) Food Avoidance, observation (24%) Low Sugar (20%) GFCF (20%) Feingold (19%)
Hyperactivity	Feingold (45%) Low Sugar (43%) GFCF (22%) Food Avoidance, observation (18%) Ketogenic (14%)
Irritability	Feingold (38%) Ketogenic (29%) Low Sugar (23%) Food Avoidance, observation (23%) GFCF (20%)
Language/Communication	Ketogenic (29%) GFCF (25%) Gluten-free (15%) Feingold (14%) Specific Carbohydrate Diet (14%)
Lethargy	Ketogenic (19%) Food Avoidance, observation (10%) Paleo (10%) Healthy (5%) Low Sugar (5%)
OCD	Paleo (10%) Ketogenic (10%) Feingold (7%) Food Avoidance, IgG/IgE (6%) Low Sugar (6%)
Reflux/Vomiting	Food Avoidance, observation (16%) Food Avoidance, IgG/IgE (13%) Paleo (10%) Ketogenic (10%) GFCF (9%)
Seizures	Ketogenic (19%) Paleo (5%) Specific Carbohydrate Diet (3%) GFCF (2%) Soy-free (2%)
Self-injury	Paleo (10%) Ketogenic (10%) Feingold (8%) GFCF (7%) Low Sugar (5%)
Sensory sensitivity	Feingold (22%) GFCF (19%) Food Avoidance, observation (17%) Paleo (14%) Food Avoidance, IgG/IgE (13%)
Sleep (falling)	Feingold (19%) Ketogenic (19%) Low Sugar (11%) GFCF (11%) Food Avoidance, observation (11%)
Sleep (staying)	Feingold (15%) Ketogenic (14%) Food Avoidance, observation (11%) Low Sugar (10%) GFCF (10%)
Social Interaction and Understanding	Ketogenic (29%) GFCF (22%) Specific Carbohydrate Diet (14%) Feingold (12%) Food Avoidance, observation (11%)
Stimming/Perseveration/Desire for Sameness	GFCF (19%) Ketogenic (19%) Specific Carbohydrate Diet (14%) Food Avoidance, observation (13%) Feingold (11%)
Tics	Paleo (10%) GFCF (6%) Food Avoidance, IgG/IgE (4%) Casein-free (4%) Feingold (3%)

**Table 6 jpm-13-01448-t006:** Top Symptom Improvements for Each Diet with at Least 10% of Respondents Improving.

Therapeutic Diets	Top Symptom Improvement (% of Participants with Improvement)
Healthy Diet	General benefit 67% Health 24% Constipation 17% Attention 12% Cognition 12% Irritability 12% Anxiety 11% Hyperactivity 11% Diarrhea 10% Sleep (falling) 10%
Feingold Diet	Hyperactivity 45% General benefit 41% Irritability 38% Attention 37% Aggression/Agitation 34% Anxiety 28% Cognition 28% Sensory sensitivity 22% Health 19% Sleep (falling) 19% Sleep (staying) 15% Constipation 14% Language/Communication 14% Social Interaction and Understanding 12% Eczema/Skin problem 11% Stimming/Perseveration/Desire for Sameness 11%
Food Avoidance Diet, Based on IgG/IgE Testing	General benefit 43% Eczema/Skin problem 22% Health 15% Reflux/vomiting 13% Sensory sensitivity 13% Anxiety 11% Attention 11% Cognition 11% Constipation 11% Hyperactivity 11% Irritability 11%
Low Sugar Diet	Hyperactivity 43% General benefit 39% Attention 25% Aggression/Agitation 23% Irritability 23% Health 20% Anxiety 18% Cognition 18% Sensory sensitivity 11% Sleep (falling) 11% Language/Communication 10% Sleep (staying) 10%
GFCF Diet	General benefit 39% Attention 31% Cognition 29% Language/Communication 25% Diarrhea 22% Hyperactivity 22% Social Interaction and Understanding 22% Eczema/Skin problem 20% Health 20% Irritability 20% Constipation 19% Sensory sensitivity 19% Stimming/Perseveration/Desire for Sameness 19% Aggression/Agitation 17% Anxiety 15% Sleep (falling) 11% Sleep (staying) 10%
Food Avoidance Diet, Based on Observation	General benefit 32% Health 24% Irritability 23% Attention 22% Eczema/Skin problem 22% Aggression/Agitation 21% Diarrhea 21% Cognition 18% Hyperactivity 18% Constipation 17% Sensory sensitivity 17% Anxiety 16% Reflux/vomiting 16% Stimming/Perseveration/Desire for Sameness 13% Language/Communication 12% Depression 11% Sleep (falling) 11% Sleep (staying) 11% Social Interaction and Understanding 11% Lethargy (easily tired) 10%
Corn-Free Diet	General benefit 44% Diarrhea 26% Constipation 22% Aggression/Agitation 17% Anxiety 11% Cognition 11% Eczema/Skin problem 11% Hyperactivity 11%
Specific Carbohydrate Diet	General benefit 57% Attention 24% Cognition 22% Anxiety 19% Diarrhea 19% Health 16% Irritability 16% Language/Communication 14% Social Interaction and Understanding 14% Stimming/Perseveration/Desire for Sameness 14% Aggression/Agitation 11% Hyperactivity 11% Sensory sensitivity 11%
Casein-Free Diet	General benefit 28% Cognition 17% Attention 16% Constipation 16% Aggression/Agitation 14% Diarrhea 14% Eczema/Skin problem 13% Health 13% Language/Communication 13% Hyperactivity 11% Anxiety 10% Irritability 10% Social Interaction and Understanding 10%
Soy-Free Diet	General benefit 45% Diarrhea 16% Health 11% Aggression/Agitation 10% Irritability 10%
Paleo Diet	General benefit 57% Attention 19% Constipation 19% Irritability 19% Anxiety 14% Cognition 14% Diarrhea 14% Health 14% Sensory sensitivity 14% Aggression/Agitation 10% Depression 10% Eczema/Skin problem 10% Hyperactivity 10% Language/Communication 10% Lethargy 10% OCD 10% Reflux/vomiting 10% Self-injury 10% Sleep (falling) 10% Sleep (staying) 10% Social Interaction and Understanding 10% Stimming/Perseveration/Desire for Sameness 10% Tics/abnormal movements 10%
Ketogenic Diet	Attention 43% General benefit 38% Cognition 38% Anxiety 33% Irritability 29% Language/Communication 29% Social Interaction and Understanding 29% Constipation 24% Lethargy 19% Sleep (falling) 19% Seizures 19% Aggression/Agitation 19% Health 19% Stimming/Perseveration/Desire for Sameness 19% Depression 14% Diarrhea 14% Eczema/Skin problem 14% Hyperactivity 14% Sleep (staying) 14% OCD 10% Reflux/vomiting 10% Sensory sensitivity 10% Self-injury 10%
Gluten-Free Diet	General benefit 36% Attention 18% Cognition 18% Diarrhea 18% Irritability 16% Constipation 15% Language/Communication 15% Aggression/Agitation 12% Health 12% Anxiety 11% Eczema/Skin problem 11% Social Interaction and Understanding 11% Stimming/Perseveration/Desire for Sameness 11% Hyperactivity 10%

## Data Availability

The data from this study are available on request from the corresponding authors.

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
