# Peer review of "Ratings of the Effectiveness of 13 Therapeutic Diets for Autism Spectrum Disorder: Results of a National Survey"

_jpm, 2023, doi:10.3390/jpm13101448_

Round 1

Reviewer 1 Report

The introduction section is too long! it could be revised

More discreption and details of statistical analyses are needed in the materials and methods section

Author Response

We thank the reviewers for their helpful comments.

Thank you for the suggestion, we have shortened parts of the introduction.

Details of statistical analysis has been added to the end of the methods section. 

We have a document with revisions in edit mode (jpm-2625306_JMJBAedits) and a cleaned up final version with citations added and updated as Zotero needed me to accept changes before I could correct the references (jpm-2625306_updatednewrefer). However, it only let's me upload one, so I'm attaching the final one. If you'd like the version with edits, please let me know how I can get it to you.

Reviewer 2 Report

Thank you for the opportunity to review this paper. The study was designed to gain an understanding of the benefits and adverse effects of therapeutic diets for individuals with autism spectrum disorder, as rated by caregivers of children and adults with ASD. The study is very timely and focuses on a really important issue. There is, however, issues that must be resolved before the study can be accepted for publication.

- „Current treatment options include behavioral therapy,  other therapies, special education, psychiatric/seizure medications, nutraceuticals, gastro intestinal treatments, and therapeutic diet”s – add references.

- At the end of the introduction, present the aims of the research (not in the material and methods).

- Mark the research questions/ hypotheses at the end of the introduction.

- Add inclusion and exclusion criteria.

- Just one piece of information that this is part of a larger study is enough, there is no need to repeat it several times.

- Line 328: "Survey data was analyzed through SPSS software" - provide more information about SPSS

- Add the strengths of a study and the limitations.

Author Response

We thank the reviewers for their helpful comments.

Current treatment options include behavioral therapy,  other therapies, special education, psychiatric/seizure medications, nutraceuticals, gastro intestinal treatments, and therapeutic diet”s – add references.

  • References have been added

At the end of the introduction, present the aims of the research (not in the material and methods).

  • This sentence was moved to the end of the introduction “The study was designed to gain an understanding of the benefits and adverse effects of therapeutic diets for individuals with autism spectrum disorder, as rated by caregivers of children and adults with ASD (and some individuals with ASD).”

Mark the research questions/ hypotheses at the end of the introduction.

  • We have added the following to the end of the introduction to address this point:  “The research questions were, will therapeutic diets offer Overall Benefit and symptom improvements for individuals with ASD, and will different diets help different symptoms?

Add inclusion and exclusion criteria.

  • The inclusion/exclusion criteria have been added to the methods section.

Just one piece of information that this is part of a larger study is enough, there is no need to repeat it several times.

  • We have removed one of the references.

Line 328: "Survey data was analyzed through SPSS software" - provide more information about SPSS

  • We added more specific information on the specific software product we used.

Add the strengths of a study and the limitations.

  • We added strengths and changed the section to "Strengths and Limitation"

We are uploading the cleaned up version with revisions. We were unable to add citations without accepting some of the edits so the most up-to-date revision does not show edits. However, we have the document in edit mode as well. We will upload the most up-to-date version. And we can send over the edited version with changes if you can let us know how to get it to you. I will try emailing it to the editor, Marilyn Zhang. 

Round 2

Reviewer 2 Report

I thank the authors for their hard work on the comments and wish them success in their future research.